# Psychometric development and evaluation of a COVID-19 social stigma scale in Indonesia

**Neti Juniarti** *, **Raini Diah Susanti, Desy Indra Yani, Nurani Nurhasanah**

Department of Community Health Nursing, Faculty of Nursing, Universitas Padjadjaran, Bandung, Indonesia

* neti.juniarti@unpad.ac.id

## Abstract

Stigma remains a significant problem globally, creating barriers to services for individuals in need, regardless of access to services. The stigma of COVID-19 primarily happened because it is a new disease with several unknowns, and these unknowns generate fear. This study aimed to conduct a psychometric development and evaluate the Public COVID-19 Stigma Scale that follows the Indonesian community's cultural background. This study used research and development design to measure the COVID-19 stigma through six steps that include seven dimensions and is culturally sensitive, starting from a literature review through to psychometric evaluation. This study was community based and was conducted in 26 regions in the Sumedang Regency. The research and development step ran from July 2021 to November 2022, with a total of 1,686 respondents. The results showed that the social stigma scale for COVID-19 consisted of 11 valid and reliable items that were separated into seven dimensions: social distancing (1 item), traditional prejudice (7 items), exclusionary sentiments (2 items), negative affect (2 items), treatment carryover (1 item), disclosure carryover (2 items), and perception of dangerousness (1 item). Further research needs to be conducted to examine the level of stigma and determine interventions to overcome the social stigma around COVID-19 in the community.

## Introduction

COVID-19 is a disease caused by the novel coronavirus (2019-nCoV). On February 11, 2020, the World Health Organization (WHO) named the disease caused by 2019-nCov, coronavirus disease, or COVID-19. The Indonesian National Task Force for COVID-19 has made various massive, systematic, and comprehensive efforts to prevent more widespread transmission of COVID-19. However, this government effort is often hampered by the negative stigma in society about the COVID-19 disease. Stigma causes people to hide their symptoms, avoid medical examinations until their condition is truly severe, and not cooperate in efforts to trace positive contacts [1]. Stigma can cause individuals to feel ashamed of their illness, feel isolated and shunned by those around them, withdraw from social life, and fear the disease they are suffering from [2]. Moreover, widespread stigma in society can cause a slowdown in the COVID-19 response process, so efforts to reduce stigma must also be the main point of attention and the central point of action to overcome the COVID-19 pandemic [3, 4].

---

**Data Availability Statement:** All relevant data are within the manuscript and its Supporting information files.

**Funding:** This study was funded by the Indonesian Ministry of Education, Cultural, and Research

---

under the "Basic Research Higher Education" scheme. The funders had no role in study design, data collection and analysis, decision to publish, or preparation of the manuscript.

**Competing interests:** The authors have declared that no competing interests exist.

Stigma remains a significant problem globally, as well, creating barriers to services for individuals in need, regardless of access to services [5]. The stigma of COVID-19 arose primarily because it is a new disease with several unknowns, and these unknowns generate fear, which translates into stigma [6]. Stigmatization is real and can negatively affect the general public response and, specifically, populations of people seeking and accessing needed care [7].

When there is widespread stigma, several problems can arise because people will tend to hide their disease or related aspects, such as a history of travel or contact with someone who was infected. Such avoidances would, in turn, delay the seeking of health care or even discourage people from practicing healthy behaviors [6]. The stigma associated with COVID-19 can lead to stress, anxiety, and depression in affected individuals. Therefore, the critical challenge with COVID-19 is combating the fear, rumors, and stigma [8]. People with stigmatized health problems may feel too embarrassed to seek treatment and may also result in those with the disease having fewer opportunities [8]. In addition to delaying medical care, patients who perceive they are stigmatized because of the disease may find that others become afraid of anyone believed to be sick, leading to prejudice against entire populations, and in some cases, stigmatization has led to violence against individuals and groups [7]. Stigma can also manifest in overt behaviors, such as avoidance, social exclusion, gossip, and harassment of an infected person [9].

When applied to health conditions, stigma can contribute to a failure to recover, lack of resources, and self-devaluation [8]. Furthermore, classifying specific individuals as fit for work based on immunological status could result in worsening stigma. The stigma is related to the inability to resume normal activities and creates resentment at the stratification of society according to "immunological fitness" [10].

Stigma is a complex process that results from the interaction of stereotypes, prejudices, and discrimination [8]. Social, community, or cultural stigma research attempts to map the stigma in society to identify the prejudice and discrimination in time and place within the community [11]. Despite various scales, measurement tools, and interventions to minimize social stigma, it nonetheless consistently exists in the community [11]. A study in Bangladesh used the stigmatized attitudes questionnaire with a five-point Likert scale that is based on guidelines for conducting behavioral insights into COVID-19 recommended by the WHO [12], with adaptations made for the specific country context. Similarly, a study in Thailand used a 10-item COVID—Public Stigma Scale (COVID-PSS) consisting of a three-factor structure: stereotype, prejudice, and fear in the community. Although the results showed that the scale was reliable, testing for the appropriateness of the 10-item COVID-PSS scores in different countries and settings is warranted to establish generalizability of the measurement [13].

Even though the COVID-19 pandemic is transitioning from a hyperacute phase to an endemic phase [14], it is still causing a significant burden on healthcare services worldwide and having profound economic and social consequences [15]. Two general challenges related to stigma of COVID-19 among the population for the upcoming years are 1) the appearance of new variants of the virus and repeated waves of local infections in an endemic phase; 2) the large numbers of patients who recovered from COVID-19 and suffering from long COVID syndrome will also be affected by different health burdens and require appropriate treatment and care [16] because there is still a mystery about the nature of SARS-COV-2 and its ability to transform in the future [15].

To date, efforts to reduce the stigma of COVID-19 around the world have been sub-optimal due to the absence of a specific measurement tool for COVID-19 stigma and the existence of various cultural and religious differences. These factors mean that stigma handling for COVID-19 has to be carried out under local cultural conditions. A tool for measuring stigma specific to COVID-19 already exists [12, 13], but it does not cover aspects of cross-cultural

adaptation, particularly in the Indonesian setting. Thus, this study's research questions were: "How can we measure COVID-19 social stigma within the cultural environment of the Indonesian community? What are the validity and reliability measures of the COVID-19 social stigma scale?" This study aimed to conduct psychometric development and evaluate the COVID-19 social stigma scale within the Indonesian community's cultural context.

## Methods

### Design of study

This research and development study focused on measuring the COVID-19 stigma with a six-step procedure that includes seven dimensions and is culturally sensitive, which is shown in the research procedure scheme (Fig 1). Each stage has a different study design and purpose that will be described in detail in a separate section. This was done in an effort to develop instrument items that can effectively measure the scale of COVID-19 stigma in the community.

### Settings

This community-based study was conducted in 26 sub-districts in the Sumedang Regency, West Java, Indonesia and ran from July 2021 to November 2022. The Sumedang Regency is one of regencies in West Java Province. The regency covers an area of 1,558.72 km$^2$ and had a population 1,159,346. The regency located within the Bandung Metropolitan area, mostly consist of rural areas. It consisted of six activity steps with different methods and characteristics of participants at each stage, which will be explained in the discussion of each step.

**Participants.** The selection of participants for the qualitative phase was carried out through a snowball technique among COVID-19 survivors and COVID-19 Task Force officers. A total of 52 informants were recruited representing all 26 sub-districts in the Sumedang Regency. The participants included 26 COVID-19 survivors and 26 COVID-19 Task Force officers. The researchers conducted interviews to explore the experiences of survivors and COVID-19 Task Force officers regarding the COVID-19 stigma in their respective regions. Interviews were conducted for 30–45 minutes face-to-face (n = 42) and by telephone (n = 10). Interviews were conducted in the Indonesian language, beginning with explaining the research objectives and securing informed consent. The researcher ensured there was no coercion in the data collection process so that participants could choose to participate in the interview voluntarily or refuse the interview.

The participants for the quantitative stage were recruited through an open online advertisement. Participants who were interested to involve in the study were filled in an online questionnaire. A total of 1,689 participants filled in the online questionnaire.

### Ethical considerations

This research upholds ethical principles, and prior to data collection, the researcher explained to potential participants the research objectives and reminded participants that they had full rights to decide whether to voluntarily participate in the research. The researcher guaranteed

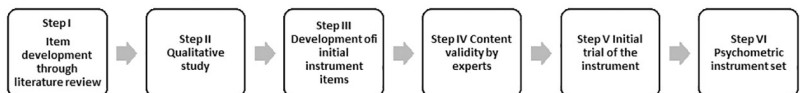

**Fig 1. Steps in the development and evaluation of a COVID-19 social stigma scale for Indonesia.**

the confidentiality of participant data identity by giving each participant a code. A consent was obtained from online and offline participants. This study did not involve minors as participants. Further, the research data were stored on a computer system with access limited to the research team only. The researchers also ensured that participants were kept physically and psychologically safe. All participants were treated fairly, regardless of different backgrounds. This study received approval from the Research Ethics Committee of Universitas Padjadjaran, Bandung, Indonesia, number 474.

## Protocol

This study was conducted in six steps, from literature review to psychometric instrument testing. The details of each step are as follows.

## Step I: Item development through literature review

The first step is a literature review to determine the theoretical concept of stigma. The literature review was conducted using the CINAHL database with the following keywords: (Stigma OR Discrimination) AND (COVID-19 OR "Corona Virus"). Inclusion criteria included publication in a peer-reviewed journal, full text available, and related to stigma problems. A total of 393 articles were obtained and analyzed using the VOS Viewer application. It was identified that there were seven dimensions of social stigma [11], as follows:

1. Social distance is when someone tries to avoid COVID-19 patients.

2. Traditional prejudice is when someone labels people with COVID-19 as less valuable.

3. Exclusionary sentiments indicate a desire to separate people with COVID-19 from others or deny those with COVID-19 certain rights.

4. Negative affect refers to emotional reactions such as disgust or hatred towards people with COVID-19.

5. Treatment carryover is when people are afraid of those who have suffered from COVID-19. This is perceived as the need to keep a secret history of having suffered from COVID-19 even though the person has recovered.

6. Disclosure carryover is when people are afraid of the reaction of others if they are found to be suffering from COVID-19.

7. Perceptions of dangerousness refers to the idea that people with COVID-19 are a risk to society.

## Step II: Qualitative study

**Data collection.** The literature review was followed by qualitative data collection using the Leiden Ethnosystem approach to explore the perspective of the stigma dimension based on the experiences of the COVID-19 Task Force and COVID-19 survivors.

Qualitative data were collected via semi-structured interviews with COVID-19 survivors and the COVID-19 Task Force by implementing health protocols during interviews to minimize the transmission of COVID-19.

**Trustworthiness.** This study focused on credibility through the re-verification of participants to ensure the accuracy of the data. The researchers provided dependability for data transcription, data reduction, and conclusions. Confirmability was accomplished by recording and making field notes systematically. Transferability was handled by ensuring that participants

met the criteria and objectives of the study and observing the principles that the collected data were saturated. Thus, data triangulation was carried out during the interactive cycle until the final stage of verification and conclusion. After the data were collected, the data analysis began with data reduction in the form of selection for simplification and focus on information obtained with coding to determine the theme. The last stage is verification or concluding in the form of meanings that are explored in depth and thoroughly to describe the experiences and perceptions of survivors and the COVID-19 Task Force regarding the COVID-19 stigma.

### Step III: Development of initial instrument items

In this step, the researcher integrated the theoretical concept of the literature review results and the themes from the qualitative study results to create a set of instrument items to measure COVID-19 stigma. The data from the literature review and the qualitative study were then used as the basis for researchers to develop instrument items. After the results were combined, seven dimensions related to COVID-19 stigma were obtained: 1) social distancing, 2) traditional prejudice, 3) exclusionary sentiments, 4) negative affect, 5) treatment carryover, 6) disclosure carryover, and 7) perceptions of dangerousness. Each dimension had a different number of items. The social distancing dimension had 32 items, the traditional prejudice dimension consisted of 15 items, the exclusionary sentiments dimension had 14 items, the negative affect dimension consisted of 11 items, the treatment carryover dimension comprised 4 items, the disclosure carryover dimension had 16 items, and the perceptions of dangerousness dimension consisted of 31 items. Thus, the initial instrument item set contained 123 statement items answered on a Likert scale with selections of "strongly agree," "agree," "don't know," "disagree," or "strongly disagree."

### Step IV: Content validity assessed by experts

The instruments compiled in the third step were now tested for content by experts. The research team selected the experts based on nursing, public health, and psychology expertise. In addition, these experts are known to have good clinical practice experience and competence in translating knowledge about social behavior theory, measurement, and evaluation. The researchers corresponded via email and personal chat with the experts. Six experts participated: two in the field of nursing, two in the field of public health, and two in psychology. They were asked to rate the instrument items based on fitness for purpose, consistency, redundancy, and clarity of statements on a "yes" and "no" dichotomy scale. They were also asked to provide comments and suggestions for improvement. The experts' input was used to revise the set of instrument items before proceeding to the quantitative trial step.

### Step V: Initial trial of the instrument

As an initial trial of the instrument to assess the performance of psychometry, a pilot study was conducted using an online survey with 113 respondents. The results of the content test and psychometric analysis were then revised according to the input before the second stage of the psychometric test was carried out on several different respondents.

### Step VI: Psychometric instrument set

After the pilot study was carried out and analyzed, followed by revision of the instrument item set, the final step was a psychometric test. A total of 1,689 respondents were used measure the COVID-19 stigma instrument based on seven dimensions so that a valid and reliable last instrument item set was obtained. Respondents in this survey was 17 years old and above.

According to Indonesian law Number 24 Year 2013, people aged 17 years old are considered adult and can have their own national identity card.

## Data analysis

Data analysis of the COVID-19 stigma instrument varied according to the research method at each stage. Data analyses were carried out thematically and interactively for qualitative research. The research team analyzed the data with the interactive method [2]. Interactive analysis is a cycle that compares the components of data reduction, data display, and verification since data collection. In this analysis cycle, researchers can freely move continuously on each of the data reduction, display, and verification elements.

For the quantitative stage, psychometric data analyses used the RASCH model.

## Results

After going through several stages of instrument item set development, the fifth step involved the initial trial of the instrument item set on 1,689 respondents. Instruments were distributed and filled out online and via social media. This section explicitly reports the findings from the sixth step.

The psychometric measurement of the revised instrument item set was carried out with 1,689 respondents. Inclusion criteria included participants age 17 years and older, including both COVID-19 survivors and non-COVID-19 survivors. Table 1 shows the demographic characteristics of respondents measuring the seven-dimension COVID-19 stigma instrument.

Of the 1,686 respondents, most respondents were women aged 17 to 30 years, were high school graduates, worked as entrepreneurs, and had not been exposed to COVID-19. Table 2 shows the seven dimensions of the COVID-19 stigma instrument after data analyses were conducted.

Cronbach's alpha value measures reliability, namely the interaction between the person and the item. Cronbach's Alpha value in the COVID-19 stigma instrument is +0.96, with a very good category. The value of person reliability is +0.94, and item reliability is +0.94. It can be concluded that the consistency of the answers from the respondents is excellent.

INFIT MNSQ and OUTFIT MNSQ, had average values for the person of 0.99, and the ideal weight was 1.00 (the closer to 1.00, the better). For INFIT ZSTD and OUTFIT ZSTD, the average value in the person was 0.99 and 0.88, where the ideal value is 0.0 (the closer the value to 0.0, the better the quality).

After the analysis, the final set of instrument items for measuring COVID-19 social stigma on a scale with seven dimensions consisted of 11 items that were infit and do not have any bias, as shown in Table 3.

## Discussion

This study develop COVID-19 social stigma scale that consist of 11 statements using seven dimensions of social stigma namely social distance, traditional prejudice, exclusionary sentiments, negative affect, treatment carryover, disclosure carryover, and perceptions of dangerousness [11]. The current study found different aspects that were found in a study in Thailand used a 10-item COVID—Public Stigma Scale (COVID-PSS) consisting of a three-factor structure: stereotype, prejudice, and fear in the community [13]. The current study found that social distancing is practiced among society. Social stigma toward those who are segregated in association with the COVID-19 patients who requires isolation; terms such as quarantine and social distancing have become a part of the common vocabulary [17].

**Table 1. Characteristics of COVID-19 stigma respondents (n = 1,689).**

| Characteristics | Frequency | Percentage |
|---|---|---|
| **Gender** | | |
| Male | 640 | 38 |
| Female | 1,046 | 62 |
| **Age** | | |
| 17–30 years old | 1,100 | 65.2 |
| 31–50 years old | 524 | 31.1 |
| > 50 years old | 62 | 3.7 |
| **Education** | | |
| No school | 6 | 4.0 |
| Elementary school | 74 | 4.4 |
| Junior high school | 223 | 13.2 |
| Senior high school | 912 | 54.1 |
| College | 471 | 27.9 |
| **Occupation** | | |
| Homemaker | 371 | 22.0 |
| Student | 350 | 20.8 |
| Teacher | 135 | 8.0 |
| Civil servant | 76 | 4.5 |
| Entrepreneur | 383 | 22.7 |
| Labor | 91 | 5.4 |
| Other jobs | 280 | 16.6% |
| **Exposure to COVID-19** | | |
| Yes | 475 | 28.8 |
| No | 1.211 | 71.8 |
| **Total** | 1,686 | 100% |

Stigma can be seen as a stereotype resulting from individuals' behaviors toward others [18]. Stigma and discrimination can arise due to ignorance about the mechanism of disease transmission. Ensuring that health-care workers have adequate knowledge about a disease is fundamental to minimizing any stigma associated with that disease [19].

Stigma is not only directed toward the patients infected but also toward people recovered from COVID-19 or those who have been released from home isolation or quarantine and even toward frontline workers involved with COVID-19 patients [17]. This current study also found similar findings for the treatment carry over and disclosure carryover. The fear of loss of financial security have led to increased anxiety, stress, and depression [17].

Stigma is also related to the perception that infected people will put others at risk of infection. Many patients are not received or treated well by their health-care workers, friends, and family members [20]. The domain of stigma is a negative label for people who are stigmatized. Stigma consists of fear, pity, needing help, avoiding, blaming, anger, separation, being dangerous to others, and coercive actions taken toward the patient [18]. This negative stamp needs to be explained to the community so that these stigma domains no longer occur.

There are several types of stigma: social or public stigma, perceived stigma, secondary stigma, self-stigma (stigma for oneself), experienced stigma, and structural stigma [21]. Of these various types, social stigma and structural stigma require serious attention. Social stigma is a negative attitude, belief, and label within the community or general public toward individuals with a disease. This social stigma against COVID-19 sufferers can lead to social isolation

**Table 2. Psychometric properties of the COVID-19 social stigma scale at the item level (N = 1,686).**

| Item number | Infit MNSQ | Outfit MNSQ | Item Measure | DIF contrast across gender | DIF contrast across educational backgrounds | DIF contrast across survivor/non-survivor | Status |
|---|---|---|---|---|---|---|---|
| **Item 1 (Social Distance [SD1])** | 1.39 | 1.51 | .49 | .273 | .0077 | .0025 | Misfit |
| People don't want to go past the houses of people with COVID-19. | | | | | | | Education bias |
| **Item 2 (SD2)** | 1.23 | 1.29 | −.15 | .2170 | .0507 | .0475 | Misfit |
| People who are exposed to COVID-19 feel shunned. | | | | | | | |
| **Item 3 (SD3)** | 1.14 | 1.17 | −.65 | .0004 | .0556 | .0010 | Infit |
| People are afraid that other people will be exposed to COVID-19. | | | | | | | Gender bias |
| **Item 4 (SD4)** | 1.00 | .97 | −.13 | 1.0000 | .1118 | .2329 | Infit |
| People close themselves off because they are afraid of being blamed by others. | | | | | | | |
| **Item 5 (Traditional Prejudice [TP1])** | .96 | .91 | .01 | .1407 | .7244 | .0028 | Infit |
| People still want to cover up their status of being exposed to COVID-19. | | | | | | | Survivor bias |
| **Item 6 (TP2)** | .95 | .97 | −.23 | .0174 | .6605 | .4783 | Infit |
| COVID-19 sufferers are afraid of their identity being shared with the public. | | | | | | | |
| **Item 7 (TP3)** | 1.03 | 1.02 | −.06 | .3379 | .0001 | .0000 | Infit |
| Many people are positive for COVID-19 but don't tell others. | | | | | | | Education, survivor biases |
| **Item 8 (TP4)** | 1.19 | 1.28 | .28 | .0009 | .0000 | .0000 | Misfit |
| Store seller refuses money from COVID-19 survivors. | | | | | | | Gender bias Education bias |
| **Item 9 (Exclusionary sentiments [ES1])** | .97 | .95 | −.02 | .2497 | .7075 | .4281 | Infit |
| People don't want to report being exposed to COVID-19 because they are ashamed and afraid of being shunned. | | | | | | | |
| **Item 10 (TP5)** | .89 | .86 | −.05 | .0853 | .5916 | .1310 | Infit |
| People feel ashamed when diagnosed with COVID-19. | | | | | | | |
| **Item 11 (ES2)** | .88 | .83 | −.07 | .3705 | .5980 | 1.0000 | Infit |
| People won't admit they are infected with COVID-19 because they are afraid of losing their jobs. | | | | | | | |
| **Item 12 (ES3)** | 1.06 | 1.09 | −.50 | .0000 | .1742 | .0175 | Infit |
| People feel paranoid about COVID-19. | | | | | | | Gender bias |
| **Item 13 (ES4)** | .85 | .81 | −.21 | .0066 | .0065 | .0146 | Infit |
| Many people have COVID-19 symptoms but don't want to report it. | | | | | | | Education, gender biases |
| **Item 14 (Negative affect [NA1])** | .96 | .92 | .14 | .6036 | .4888 | .6032 | Infit |
| People still feel that suffering from COVID-19 is a disgrace. | | | | | | | |
| **Item 15 (NA2)** | 1.16 | .92 | .69 | .0025 | .0000 | .0025 | Infit |
| People affected by COVID-19 are considered disgusting. | | | | | | | Gender bias Education, survivor biases |
| **Item 16 (Treatment Carryover [TC1])** | 1.27 | 1.24 | .89 | .0002 | .0000 | .3009 | Misfit |
| A recovered COVID-19 survivor was rejected at local shops while shopping. | | | | | | | Bias gender Bias Education |

*(Continued)*

**Table 2.** (Continued)

| Item number | Infit MNSQ | Outfit MNSQ | Item Measure | DIF contrast across gender | DIF contrast across educational backgrounds | DIF contrast across survivor/non-survivor | Status |
|---|---|---|---|---|---|---|---|
| **Item 17 (TC2)** | 1.04 | 1.00 | .17 | 1.0000 | .0000 | .0062 | Infit |
| COVID-19 survivors feel awkward and afraid when meeting people in public. | | | | | | | Education, survivor biases |
| **Item 18 (Disclosure Carryover [DC1])** | 1.06 | 1.03 | .35 | .0243 | .0033 | 1.0000 | Infit |
| Even though self-isolation has been completed, there are still people who stay away from former COVID-19 sufferers. | | | | | | | Gender, Education biases |
| **Item 19 (DC2)** | .84 | .82 | −.15 | 1.0000 | .0001 | .1451 | Infit |
| People don't want to do a COVID-19 test if they feel symptoms. | | | | | | | Education bias |
| **Item 20 (DC3)** | 1.00 | 1.02 | −.47 | .3678 | .0000 | 1.0000 | Infit |
| People prefer to self-isolate without doing COVID-19 checks. | | | | | | | Education bias |
| **Item 21 (DC4)** | .88 | .84 | −.31 | .0556 | .0443 | .7365 | Infit |
| People don't want to do a rapid test for fear of a positive result. | | | | | | | |
| **Item 22 (DC5)** | .86 | .88 | −.01 | .0471 | .0655 | .2457 | Infit |
| People are secretly dealing with COVID-19. | | | | | | | Gender bias |
| **Item 23 (DC6)** | .84 | .82 | .00 | .0163 | .2324 | .2824 | Infit |
| People checked themselves into private clinics but did not report to the public health center. | | | | | | | Gender bias |
| **Item 24 (Treatment carryover [TC3])** | .81 | .79 | −.04 | .6579 | .4589 | .2582 | Infit |
| People hide their status of being exposed to COVID-19 for fear of losing their jobs. | | | | | | | |
| **Item 25 (DC7)** | .91 | .88 | −.02 | 1.0000 | .1368 | .2816 | Infit |
| People prefer self-isolation without being tested, but they do not commit to doing self-isolation properly. | | | | | | | |
| **Item 26 (Negative affect [NA3])** | .83 | .83 | .04 | 1.0000 | .6537 | 1.0000 | Infit |
| People with symptoms of anosmia don't want to do swab tests for fear of being disgraced. | | | | | | | |
| **Item 27 (Perception of dangerousness [PD1])** | .86 | .88 | .01 | .4113 | .1193 | 1.0000 | Infit |
| People don't want to mention their close contacts when tracing for fear of losing other people's sustenance. | | | | | | | |
| Mean | .99 | .99 | .00 | | | | |
| SD | .15 | .18 | .33 | | | | |

of patients, negative prejudices toward them, exclusion from the community, and other negative effects. Even when they have recovered, those with COVID-19 are still considered to bring disease and are thought to be dangerous to society [11].

The other stigma that must be considered is the structural stigma carried out by an agency, company, or law that rejects sick people. This includes social conditions, cultural norms, and institutional practices that limit the stigmatized population's opportunities, resources, and well-being. Both types of stigma can have an impact on social isolation and lead to the failure to seek care to cope with the disease [21]. For patients with COVID-19, these two stigmas can cause them to not seek medical attention and treatment when they have COVID-19 symptoms,

**Table 3. The seven-dimensional COVID-19 social stigma scale.**

| No. | Item |
|---|---|
| | **Dimension 1: Social Distancing** |
| 1 | People close themselves off because they are afraid of being blamed by others. |
| | **Dimension 2: Traditional Prejudice** |
| 2 | COVID-19 sufferers are afraid of their identity being shared with the public. |
| 3 | People feel ashamed when diagnosed with COVID-19. |
| | **Dimension 3: Exclusionary Sentiments** |
| 4 | People do not want to report being exposed to COVID-19 because they are ashamed and afraid of being shunned. |
| 5 | People would not admit they are infected with COVID-19 because they are afraid of losing their jobs. |
| | **Dimension 4: Negative Affect** |
| 6 | People still feel that suffering from COVID-19 is a disgrace. |
| 7 | People with symptoms of anosmia don't want to do swab tests for fear of being disgraced. |
| | **Dimension 5: Treatment Carryover** |
| 8 | People hide their status of being exposed to COVID-19 for fear of losing their jobs. |
| | **Dimension 6: Disclosure Carryover** |
| 9 | People do not want to do a rapid test for fear of a positive result. |
| 10 | People prefer self-isolation without being tested, but they do not commit to doing self-isolation properly. |
| | **Dimension 7: Perception of Dangerousness** |
| 11 | People do not want to mention their close contacts when tracing for fear of losing other people's friendship. |

or if they have seen a health-care provider, refusal to continue their treatment [22]. The social stigma is worsened when the health professional also had stigma. A study in Indonesia found that 21.9% respondents had stigma associated with COVID-19 during the early phase of the COVID-19 outbreak in Indonesia, the stigmatizing attitudes might have been influenced by lack of information and knowledge about the disease [23].

To reduce stigma COVID-19, there are specific recommendations for interventions that focus on: (1) language/words used in relation to COVID-19 and affected people; (2) media/journalistic practices; (3) public health interventions; (4) targeted public health interventions for key groups and (5) involving communities and key stakeholders [24]. Active engagement of Community Health Nurses (CHN) in the primary health care system plays essential tasks in managing COVID-19 through providing comprehensive services by CHNs, utilization of technology, family nursing care, community empowerment, and multi-program and multi-sector collaborations [25]; thus, can help reduce stigma of COVID 19 in the society. An integrated approach is needed to improve the knowledge of all stakeholders about the disease [26].

The main strength of this study is that it provides a comprehensive perspective from qualitative approach that involved community leader as the COVID-19 task force and COVID-19 survivors. Perspectives from these participants provided a comprehensive picture for understanding the domains of the COVID-19 social stigma. The qualitative approach followed by quantitative design to test the validity and reliability of the COVID-19 social stigma scale.

The main limitation of the study is that it is a study conducted in the region of West Java Province, which may have a different context from other regions in Indonesia, or from the wider global community. Therefore, the context may be limited to this particular region, and might not be able to be generalized to the entire population. However, the domain of social stigma scale can be used beyond the sites investigated in the study. Therefore, the COVID-19 social stigma scale that have been identified in this study might inform other researchers,

health professionals, and policy makers in different settings to make adjustments and improvements based on their own specific contexts.

## Conclusions

Overall, the social stigma scale for COVID-19 consisted of 11 valid and reliable items. The items were separated into seven dimensions: social distancing (1 item), traditional prejudice (7 items), exclusionary sentiments (2 items), negative affect (2 items), treatment carryover (1 item), disclosure carryover (2 items), and perception of dangerousness (1 item). Further research must be conducted to examine the level of stigma and find interventions to overcome the social stigma of COVID-19 in the community.

## Supporting information

**S1 Appendix. Final version the seven-dimensional COVID-19 social stigma scale.**
(DOCX)

**S1 Data set. Stigma COVID-19 research.**
(ZIP)

## Author Contributions

**Conceptualization:** Neti Juniarti, Desy Indra Yani, Nurani Nurhasanah.

**Data curation:** Desy Indra Yani, Nurani Nurhasanah.

**Formal analysis:** Neti Juniarti, Raini Diah Susanti.

**Funding acquisition:** Neti Juniarti.

**Investigation:** Desy Indra Yani.

**Methodology:** Neti Juniarti.

**Resources:** Neti Juniarti.

**Software:** Neti Juniarti.

**Writing – original draft:** Raini Diah Susanti.

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
