## [Decision Letter · Decision Letter 0]

26 Jan 2023

PONE-D-22-32944Dear Editor of Health PLOS ONE,

Full title : Psychometric Development and Evaluation of a COVID-19 Social Stigma Scale in IndonesiaPLOS ONE

Dear Dr. Juniarti,

Thank you for submitting your manuscript to PLOS ONE. After careful consideration, we feel that it has merit but does not fully meet PLOS ONE’s publication criteria as it currently stands. Therefore, we invite you to submit a revised version of the manuscript that addresses the points raised during the review process.

We look forward to receiving your revised manuscript.

Kind regards,

Muhammad Arsyad Subu, Ph.D

Academic Editor

PLOS ONE

Journal Requirements:

"This study was funded by the Indonesian Ministry of Education, Cultural, and Research under the "Basic Research Higher Education" scheme"

"NO authors have competing interests"

Reviewers' comments:

Reviewer's Responses to Questions

**Comments to the Author**

1. Is the manuscript technically sound, and do the data support the conclusions?

Reviewer #1: Yes

2. Has the statistical analysis been performed appropriately and rigorously? 

Reviewer #1: Yes

3. Have the authors made all data underlying the findings in their manuscript fully available?

Reviewer #1: Yes

4. Is the manuscript presented in an intelligible fashion and written in standard English?

Reviewer #1: Yes

5. Review Comments to the Author

Reviewer #1: The topic is very interesting. However, I keep wondering why it is still important while COVID-19 has subsided. I think stigma could be associated with COVID-19 and other communicable diseases.

Settings: more information is needed to be mentioned.

it is too late to talk about participants.

line 151: please resyntax this sentence.

line 156: I am wondering how come interview could help in observation of covid prevention protocols.

line 192: please replace don't agree with disagree.

line 215: there two data analysis sections into the article which could confuse the reader.

Discussion: it is like another introduction to the topic. I do recommend restructure it to include similar and opposed study findings.

6. PLOS authors have the option to publish the peer review history of their article (what does this mean?). If published, this will include your full peer review and any attached files.

Reviewer #1: No

---

## [Author Response · Author response to Decision Letter 0]

12 Mar 2023

Journal Requirements:

https://journals.plos.org/plosone/s/file?id=wjVg/PLOSOne_formatting_sample_main_body.pdf and https://journals.plos.org/plosone/s/file?id=ba62/PLOSOne_formatting_sample_title_authors_affiliations.pdf � Author response: Yes, has been revised

2. You indicated that you had ethical approval for your study. In your Methods section, please ensure you have also stated whether you obtained consent from parents or guardians of the minors included in the study or whether the research ethics committee or IRB specifically waived the need for their consent. � Author response: I didn’t have minors as my participants

3. Please provide additional details regarding participant consent. In the ethics statement in the Methods and online submission information, please ensure that you have specified what type you obtained (for instance, written or verbal, and if verbal, how it was documented and witnessed). If your study included minors, state whether you obtained consent from parents or guardians. If the need for consent was waived by the ethics committee, please include this information. Author response: All participants give an online consent. I didn’t have minors as my participants

Once you have amended this/these statement(s) in the Methods section of the manuscript, please add the same text to the “Ethics Statement” field of the submission form (via “Edit Submission”). � Author response: This has been added to the ethic statement

"This study was funded by the Indonesian Ministry of Education, Cultural, and Research under the "Basic Research Higher Education" scheme" � The statement was added in the acknowledgement

Please state what role the funders took in the study. If the funders had no role, please state: ""The funders had no role in study design, data collection and analysis, decision to publish, or preparation of the manuscript."" � Author response: The statement was added in the acknowledgement

Please include this amended Role of Funder statement in your cover letter; we will change the online submission form on your behalf. � Author response: The statement was added in the Cover letter

"NO authors have competing interests"

 This information should be included in your cover letter; we will change the online submission form on your behalf. � Author response: The statement was added in the Cover letter

Important: If there are ethical or legal restrictions to sharing your data publicly, please explain these restrictions in detail. Please see our guidelines for more information on what we consider unacceptable restrictions to publicly sharing data: http://journals.plos.org/plosone/s/data-availability#loc-unacceptable-data-access-restrictions. Note that it is not acceptable for the authors to be the sole named individuals responsible for ensuring data access. � The statement was added in the online submission and a supporting file of minimal dataset will be uploaded with the manuscript. 

7. Please include your full ethics statement in the ‘Methods’ section of your manuscript file. In your statement, please include the full name of the IRB or ethics committee who approved or waived your study, as well as whether or not you obtained informed written or verbal consent. If consent was waived for your study, please include this information in your statement as well. � I have included the full name of the ethic committee

8. Please include captions for your Supporting Information files at the end of your manuscript, and update any in-text citations to match accordingly. Please see our Supporting Information guidelines for more information: http://journals.plos.org/plosone/s/supporting-information. � Author response: The Captions was added

Reviewer #1: The topic is very interesting. However, I keep wondering why it is still important while COVID-19 has subsided. I think stigma could be associated with COVID-19 and other communicable diseases. � Author response: Authors added more reference about the importance of COVID-19 social stigma in the endemic phase (Line 72-80). 

Settings: more information is needed to be mentioned. � Author response: Authors added more information about the setting.

it is too late to talk about participants. � Author response: The Participants sectioned was moved right after the setting to describe participants in the qualitative and quantitative stages.

line 151: please resyntax this sentence. � Author response: the sentence was changed to “A total of 52 informants were recruited representing all 26 sub-districts in the Sumedang district” (Line 144-145)

line 156: I am wondering how come interview could help in observation of covid prevention protocols. � Author response: the phrase “to observe COVID-19 prevention protocols” was deleted as there was a translation error (line 150).

line 192: please replace don't agree with disagree. �Author response: replaced with disagree (line 185)

line 215: there two data analysis sections into the article which could confuse the reader. �Author response: The data analysis section (line 161) was combined in the data analysis section (line 215) so there is only one data analysis section.

Discussion: it is like another introduction to the topic. I do recommend restructure it to include similar and opposed study findings. � Author response: The discussion section was restructured to include similar and opposed study findings, and other related reference related to the results.

---

## [Editor Report · Decision Letter 1]

20 Mar 2023

Psychometric Development and Evaluation of a COVID-19 Social Stigma Scale in Indonesia

PONE-D-22-32944R1

Dear Dr. Juniarti,

We’re pleased to inform you that your manuscript has been judged scientifically suitable for publication and will be formally accepted for publication once it meets all outstanding technical requirements.

Kind regards,

Muhammad Arsyad Subu, Ph.D

Academic Editor

PLOS ONE
---

## [Editor Report · Acceptance letter]

27 Mar 2023

PONE-D-22-32944R1 

Psychometric Development and Evaluation of a COVID-19 Social Stigma Scale in Indonesia 

Dear Dr. Juniarti:

I'm pleased to inform you that your manuscript has been deemed suitable for publication in PLOS ONE. Congratulations! Your manuscript is now with our production department. 

Kind regards, 

on behalf of

Dr. Muhammad Arsyad Subu 

Academic Editor

PLOS ONE